# Verifying Alignment Constraints Under Finite-Sample Uncertainty in Composite-Data Regimes

**Blossom Metevier, Max Springer, Bohdan Turbal, Aleksandra Korolova**
Center for Information Technology Policy
Princeton University
{bmetevier,maxspringer,bt4811,korolova}@princeton.edu

## Abstract

Machine learning systems are increasingly deployed in settings where failures to uphold safety or fairness can result in significant consequences. In such settings, practitioners may seek assurances that learned models satisfy safety and fairness criteria despite the statistical uncertainty induced by training with finite data. We study an existing framework for obtaining high-confidence guarantees that learned models satisfy such criteria under finite-sample uncertainty, and provide a unifying proof framework that identifies the minimal statistical conditions needed for these guarantees. We then consider settings in which the data available for training consists of a mixture of trusted data and model-inferred quantities, such as proxy labels or automated evaluations. These composite data regimes arise naturally in modern alignment pipelines, but they can violate the statistical assumptions required for high-confidence guarantees to hold. As a result, a model may only appear to satisfy safety or fairness criteria. We characterize how this failure arises, and derive sufficient conditions under which the guarantee can be recovered.[1]

## 1 Introduction

Modern machine learning (ML) systems are increasingly used in high-stakes domains such as hiring, healthcare, and finance (Valero, 2024; Bagattini et al., 2025). In these domains, stakeholders often seek assurances regarding fairness, safety, and reliability. However, in practice, providing such assurances is challenging under *finite-sample uncertainty*: with limited data, estimates of safety or fairness criteria are noisy, and using the same dataset both to optimize model performance and to verify those criteria can make them appear to hold even when they do not generalize. This can lead to *fairness washing*, where a discriminatory model is falsely certified as satisfying fairness constraints (Aïvodji et al., 2019; Cotter et al., 2019). This risk is compounded when evaluating fairness criteria across multiple groups, where multiple-comparisons effects can increase the chance of incorrectly concluding that an unfair model satisfies the relevant criteria (Rupert Jr, 2012).

Prior work addresses this by providing a framework for obtaining high-confidence guarantees that an ML system satisfies constraints on model behavior under finite-sample uncertainty, achieved by decoupling model optimization from statistical verification of those constraints (Thomas et al., 2019). This framework also supports abstention, an important safeguard in high-risk settings, by avoiding deployment when there is insufficient statistical confidence that a model satisfies the relevant criteria, such as when the available data are insufficient for verification. Building on this framework, subsequent methods have been developed across a variety of settings, including supervised learning, reinforcement learning, and contextual bandits, and instantiated with a range of fairness and safety constraints (Thomas et al., 2019; Kostas et al., 2021; Metevier et al., 2019; Chittepu et al., 2025; Weber et al., 2025).

These methods are typically analyzed under the assumption that the data used for optimization and constraint verification are drawn from a common underlying distribution. In many practical ML

---

[1]An extended version of this work is available at: https://bmetevier.github.io/papers/inferreddata.pdf.

pipelines, however, a limited set of trusted or ground-truth data, such as human annotations or expert labels, is combined with *model-inferred quantities*, that is, labels, attributes, scores, or feedback produced by other models rather than directly observed or annotated. Examples of such *composite data regimes* arise when demographic attributes are inferred from proxy information such as names, text, or profile data (Wood-Doughty et al., 2018; Hinds & Joinson, 2018), when automated toxicity classifiers are used to score model outputs (Gehman et al., 2020), or when reward models generate predicted preferences or reward estimates (Shi et al., 2024). In these composite data regimes, model-inferred quantities may be noisy or systematically biased (Lockhart et al., 2023; Borkan et al., 2019), invalidating the statistical assumptions required for prior methods to provide high-confidence guarantees. When this happens, the statistical verification of fairness criteria may certify a model as fair even when the underlying fairness constraint is violated, reintroducing the risk of fairness washing; related concerns have been observed in fairness assessment under unobserved, proxy, or noisy protected attributes (Kallus et al., 2022; Ghosh et al., 2021).

We develop a unifying proof framework that captures a broad class of prior methods for providing high-confidence guarantees under finite-sample uncertainty and shows that they share a common statistical structure (Section 3). This framework makes explicit the assumptions required for valid statistical verification and characterizes when such guarantees hold. In particular, we show that in composite data regimes, these guarantees can fail when verification relies on model-inferred quantities, but can be recovered under appropriate restrictions on the data used for verification of fairness constraints, such as restricting verification to a trusted ground-truth dataset (Section 4).

## 2 PROBLEM SETTING & BACKGROUND

We consider a setting where an algorithm optimizes a primary objective, such as minimizing loss or maximizing reward, subject to satisfying alignment constraints with high probability under finite-sample uncertainty. The algorithm has access to a dataset $D = \{d_i\}_{i=1}^n$ of $n$ samples drawn from a distribution $\mathcal{D}$ over the space of data instances $\mathcal{Z}$. We treat $D$ as a random variable taking values in $\mathcal{Z}^n$; accordingly, all probabilities and expectations are taken over the randomness in $D$, as well as internal randomness of the algorithm when applicable. The algorithm is a mapping $\mathrm{alg} : \mathcal{Z}^n \to \Theta \cup \{\bot\}$, where $\Theta$ denotes the solution space (e.g., feasible model parameters or policies), and $\bot$ denotes abstention.

### 2.1 ALIGNMENT SPECIFICATIONS

We define an *alignment specification* as a function $g : \Theta \to \mathbb{R}$, where for all $\theta \in \Theta$, $g(\theta)$ is defined on the underlying data-generating distribution $\mathcal{D}$ and calibrated such that $g(\theta) > 0$ indicates the model is misaligned (e.g., exhibits bias, unsafe behavior, or performance regression), and $g(\theta) \leq 0$ indicates it is aligned. We adopt a broad definition of alignment, viewing it as the satisfaction of deployment-specific constraints defined on the underlying data-generating distribution.

In many settings, alignment specifications depend on multiple underlying expectations. For example, fairness constraints may involve differences or ratios of conditional risks across groups (Feldman et al., 2015; Hardt et al., 2016), while safety constraints may be formulated as limits on expected costs or risks (Achiam et al., 2017). To capture such cases, we adopt a standard structural assumption from prior work (Metevier et al., 2019). In particular, we assume that an alignment specification can be expressed as a function of $d$ *base variables*, where each base variable is an expectation that admits an unbiased estimator, and these variables are combined using operations such as addition, multiplication, negation, scalar inverse, maximum, and absolute value.[2]

**Assumption 1** (Decomposable Alignment Specification). *For an alignment specification $g : \Theta \to \mathbb{R}$, there exists a function $\phi$ and a finite set of base variables $\{z_j(\theta)\}_{j=1}^d$ such that*

$$g(\theta) = \phi\big(\{z_j(\theta)\}_{j=1}^d\big), \tag{1}$$

*where each base variable $z_j(\theta)$ is a scalar defined as the expected value of an unbiased estimator $\hat{z}_j$, $z_j(\theta) \equiv \mathbb{E}[\hat{z}_j]$. Given a fixed candidate solution $\theta_c$, each estimator $\hat{z}_j$ is computed on the verification*

---

[2]Related work extends this framework to settings where unbiased estimation is infeasible, instead deriving guarantees from one-sided bounds on the relevant quantities (Satija et al., 2021; Yang et al., 2021; Metevier et al., 2019).

*dataset $D_V$ as the sample average of independent and identically distributed outputs generated by a function $\nu_j$: $\hat{z}_j = \text{Average}(\nu_j(D_V, \theta_c))$. The plug-in estimator of the alignment specification is $\hat{g} = \phi(\{\hat{z}_j\}_{j=1}^d)$.*

Group fairness in classification provides a simple illustration of this assumption. Many group fairness criteria compare predictive rates across demographic groups and therefore naturally depend on multiple conditional expectations. As one example, disparate impact (Feldman et al., 2015) is defined in terms of group-specific positive prediction rates and can be written directly in the form required by Assumption 1.

**Example 2.1 (Group Fairness in Classification).** Let $T \in \{G_1, G_2\}$ denote a sensitive attribute identifying two demographic groups, and let $\hat{Y} = f_\theta(X) \in \{0, 1\}$ denote the model prediction. Disparate impact, often operationalized via the $80\%$ rule, requires the positive prediction rate for group $G_1$ to be sufficiently large relative to that of group $G_2$. This can be written as the alignment specification

$$g(\theta) = 0.8 - \frac{\Pr(\hat{Y} = 1 \mid T = G_1)}{\Pr(\hat{Y} = 1 \mid T = G_2)}. \tag{2}$$

This specification decomposes into the base variables $z_1(\theta) = \mathbb{E}[\hat{Y} \mid T = G_1]$ and $z_2(\theta) = \mathbb{E}[\hat{Y} \mid T = G_2]$, combined via $\phi(z_1, z_2) = 0.8 - \frac{z_1}{z_2}$.

## 2.2 SELDONIAN ALGORITHMS

In practice, alignment specifications must be estimated from finite data $D$. As a result, a model that violates an alignment constraint on the underlying data-generating distribution may nevertheless appear to satisfy it when evaluated on $D$. This mismatch arises because the quantities defining alignment are estimated with statistical uncertainty, and optimization over finite data can select models that benefit from these fluctuations rather than genuinely satisfying the desired constraint (Thomas et al., 2019). Consequently, simply checking whether an empirical estimate of an alignment specification is aligned (i.e., that the corresponding constraint is non-positive) does not provide a reliable guarantee that the deployed model is aligned. This problem is compounded when alignment is defined using multiple estimated quantities, or when multiple constraints are evaluated simultaneously, since these settings further increase the chance of spuriously certifying a misaligned model (Rupert Jr, 2012). To address this risk, we study the Seldonian framework, which provides high-confidence guarantees against falsely certifying alignment under finite-sample uncertainty (Thomas et al., 2019).

We adopt the Seldonian framework's perspective of placing the guarantee on the algorithm, not any fixed model. This is due to the fact that under finite-sample uncertainty, the output of an ML system itself is random: different draws of the training dataset $D$ can lead the same algorithm to return different models. A probabilistic alignment guarantee therefore requires that, with high probability over the random draw of $D$ (and any internal randomness of the algorithm, if present), the solution returned by the algorithm satisfies the desired alignment constraints on the underlying data-generating distribution.

Formally, we say that an algorithm alg satisfies a *probabilistic alignment guarantee* for a collection of $k$ alignment constraints $\{g_\iota(\theta) \leq 0\}_{\iota=1}^k$ with high probability if for each constraint $\iota$,

$$\Pr_{D \sim \mathcal{D}^n}(g_\iota(\text{alg}(D)) \leq 0) \geq 1 - \delta_\iota, \tag{3}$$

where $\delta_\iota \in (0, 1)$ is a user-specified failure tolerance. Equivalently, if we were to rerun the full algorithm on many independent datasets drawn from $\mathcal{D}$, the returned solution would violate constraint $\iota$ on at most $\delta_\iota$ fraction of runs.

The Seldonian framework specifies a three-step protocol for achieving this guarantee, summarized in Algorithm 1. First, the **partition step** (line 1) splits $D$ into a candidate selection set $D_{CS}$ and a verification set $D_V$. Next, the **candidate selection step** (line 2) uses $D_{CS}$ to produce a candidate model $\theta_c$. Finally, in the **verification step** (lines 3–8), the algorithm uses $D_V$ to compute a high-confidence upper bound (HCUB) $U(\hat{g}_\iota)$, where $\hat{g}_\iota$ is an estimate of $g_\iota(\theta_c)$ computed from $D_V$. If

each bound certifies that the corresponding alignment constraint is satisfied, i.e., $U(\hat{g}_\iota) \leq 0$, then $\theta_c$ is returned. Otherwise, the algorithm abstains, returning $\perp$. We describe each step in more detail below.

**Partition step.** In the Seldonian framework, the input dataset $D$ is modeled as consisting of samples drawn independently and identically distributed (i.i.d.) from an underlying data-generating distribution, and Algorithm 1 partitions this dataset at random into $D_{CS}$ and $D_V$ (Thomas et al., 2019).

**Assumption 2** (I.I.D. Training Data). *The training dataset $D = \{d_i\}_{i=1}^n$ consists of samples drawn independently and identically from the underlying data-generating distribution $\mathcal{D}$.*

---

**Algorithm 1** Probabilistic Alignment Protocol

---
**Require:** Dataset $D$; Confidence levels $\{\delta_\iota\}_{\iota=1}^k$
**Ensure:** Candidate $\theta_c$ or $\perp$
1: $D_{CS}, D_V \leftarrow \text{Partition}(D)$
2: $\theta_c \leftarrow \text{CandidateSelection}(\{\delta\}_{\iota=1}^k, D_{CS})$
3: **for** $\iota \in \{1, \ldots, k\}$ **do**  ▷ Verification Step
4:     $\hat{z}_j = \text{Average}(\nu_j(D_V, \theta_c))$
5:     $\hat{g}_\iota = \phi_\iota(\{\hat{z}_j\}_{j=1}^d)$
6:     **if** $U(\hat{g}_\iota) > 0$ **return** $\perp$ **end if**
7: **end for**
8: **return** $\theta_c$

---

**Assumption 3** (Random Partitioning). *Algorithm 1 partitions $D$ into disjoint subsets $D_{CS}$ and $D_V$ using a random split that does not depend on the observed data values.*

These assumptions establish the conditions needed for the verification data to be independent of the candidate model $\theta_c$, allowing statistically valid HCUBs on the alignment constraints. We formally show this in Section 3.

**Candidate selection step.** The candidate selection step employs a domain-specific optimization on $D_{CS}$ to search for a high-utility model. While this step is crucial for performance, the alignment guarantee is governed by the statistical validity of the verification step, provided that the verification data are appropriately independent of the candidate model (formalized in Appendix C). Accordingly, we focus our analysis on the verification step, which we discuss next.

**Verification step.** The verification step evaluates whether the candidate model $\theta_c$ satisfies each alignment constraint using $D_V$. For each constraint $\iota$, the HCUB $U(\hat{g}_\iota)$ is constructed so that $\Pr(g_\iota(\theta_c) \leq U(\hat{g}_\iota)) \geq 1 - \delta_\iota$. If all constraints satisfy $U(\hat{g}_\iota) \leq 0$, the candidate is returned; otherwise, the algorithm abstains. To make this concrete, suppose $\hat{g}_\iota$ is computed as the empirical mean of $m$ i.i.d. samples from the $D_V$. A one-sided Student's $t$-test can be used to construct the upper bound:

$$U_{\text{ttest}}(\hat{g}_\iota) = \hat{g}_\iota + \frac{\hat{\sigma}_\iota}{\sqrt{m}} \, t_{1-\delta_\iota, m-1},$$

where $\hat{\sigma}_\iota$ is the sample standard deviation and $t_{1-\delta_\iota, m-1}$ is the appropriate quantile of the $t$-distribution. This yields an HCUB on the true constraint value. Appendix B provides additional constructions used in prior work.

## 2.3 ADDITIONAL RELATED WORK

**High-Confidence Constraint Satisfaction.** Beyond Seldonian algorithms, other frameworks also provide formal guarantees on model behavior. For example, conformal prediction provides distribution-free guarantees for predictive coverage, while PAC-Bayesian analysis gives generalization bounds for randomized predictors (Vovk et al., 2005; McAllester, 1999). However, these frameworks are not designed to directly certify arbitrary user-specified constraints on model behavior, such as conditional fairness metrics or domain-specific safety requirements. Our contribution is to identify structural conditions underlying prior Seldonian methods, and to study how these conditions can fail or be recovered in composite-data settings.

**Composite Data in Alignment.** Contemporary alignment pipelines often operate in composite data regimes, in which a limited set of trusted or ground-truth data is combined with model-inferred quantities such as AI-generated feedback or inferred attributes (Bai et al., 2022; Hinds & Joinson, 2018; Wood-Doughty et al., 2018). As these pipelines are deployed at increasing scale, such regimes

are likely to become more common, since fairness evaluation will often rely on model-inferred quantities when more trusted data are unavailable or too costly to obtain (Kallus et al., 2022; Hinds & Joinson, 2018). The statistical guarantees studied in the Seldonian framework are typically developed under the assumption that the data used for candidate selection and verification are drawn i.i.d. from a single underlying distribution (Thomas et al., 2019). This assumption does not neatly fit composite data regimes, where verification may rely on quantities that are noisy, biased, or drawn from multiple sources (Kallus et al., 2022). We study how this mismatch can invalidate high-confidence guarantees under finite-data uncertainty, and identify conditions under which they can be recovered.

## 3 Unifying Framework for Statistical Verification

Prior methods for achieving high-confidence alignment guarantees have been developed independently across supervised learning, reinforcement learning, and contextual bandits (Thomas et al., 2019; Kostas et al., 2021; Metevier et al., 2019). Each method proves its own guarantee under its own assumptions, making it difficult to identify what is structurally required as compared to what is setting-specific. In this section, we develop a general proof framework that unifies these methods by isolating the sufficient statistical structure needed for the alignment guarantee (equation 3) to hold. First, we identify sufficient conditions under which the HCUB is statistically valid in the standard i.i.d. setting (Theorem 1). Then we show that pre-existing alignment protocols satisfy these conditions. The full proofs are deferred to Appendix C.

### 3.1 Sufficient Conditions for the Alignment Guarantee

We now identify two requirements under which an algorithm following Algorithm 1 satisfies the probabilistic alignment guarantee in equation 3. Operationally, these requirements specify what must hold for the verification step to certify a candidate model with the stated confidence. The first is a regularity requirement on the verification estimates used by the chosen HCUB method. This requirement is method-specific and depends on the choice of HCUB; examples include boundedness, approximate normality, or finite variance. The second is a requirement that candidate selection be independent of verification. This is needed because the HCUB is constructed to provide coverage for a fixed parameter value $\theta_c$; if the data used to evaluate $\theta_c$ were also used to select it, then the verification estimates become adaptively chosen and the stated coverage guarantee may fail. Together with the decomposable structure in Assumption 1, these requirements imply the desired guarantee. We formalize these requirements as follows.

**Assumption 4** (Regularity Conditions for HCUB). *The verification estimates, $\hat{g}_\iota$, satisfy the method-specific regularity conditions required by the chosen HCUB function, $U(\cdot)$.*

For example, if the $t$-test is used, this requires that the distribution of verification scores is sufficiently symmetric. If, instead, Hoeffding's inequality is used, this requires that the per-sample verification outputs are almost surely bounded in a known interval $[a, b]$.

Next, we specify a condition on the protocol itself, specifically, that the model produced by the candidate selection step be statistically independent of the verification data.

**Condition 1** (Candidate-Verification Independence). *The candidate solution $\theta_c$ produced by the candidate selection step is statistically independent of the verification data, $D_V$.*

Unlike Assumption 4, which concerns statistical properties of the verification estimates, this requirement can be established from the design of the protocol itself. We show in Sections 3.2 and 4 how different implementations of the partition step satisfy this condition.

Combining these with the decomposable structure of the alignment specification (Assumption 1), we obtain the following result.

**Theorem 1** (Sufficient Conditions for Alignment Guarantee). *If an algorithm follows the protocol in Algorithm 1 and satisfies Assumptions 1 and 4 and Condition 1, then for all $\iota \in [k]$,*

$$\Pr_{D \sim \mathcal{D}^n} \left( g_\iota(\mathrm{alg}(D)) \leq 0 \right) \geq 1 - \delta_\iota.$$

More intuitively, the probabilistic alignment guarantee follows if the alignment specification admits a base-variable decomposition (Assumption 1); the verification estimates satisfy the regularity

conditions required by the chosen HCUB (Assumption 4); and the candidate model is statistically independent of the verification data (Condition 1). The remainder of our analysis shows how these requirements are satisfied in settings assumed by prior work (Section 3.2), and how they can fail or be recovered in composite-data regimes (Section 4).

## 3.2 Satisfying Sufficient Conditions in Prior Work

The sufficient conditions in Theorem 1 are abstract and do not yet prescribe how exactly to partition data, or what distributional assumptions need to be made. In practice, existing alignment protocols achieve Condition 1 through two concrete assumptions: i.i.d. training data and random partitioning. Under these assumptions, the independence in Condition 1 follows directly.

**Property 1** (Independence Under Standard Assumptions). *Under Assumptions 2 and 3, Condition 1 holds, i.e., $\theta_c = \mathrm{CandidateSelection}(D_{CS})$ is statistically independent of $D_V$.*

The proof uses i.i.d. factorization of the joint density and the data-independent split to show that $D_{CS}$ and $D_V$ are unconditionally independent, from which the independence of $\theta_c$ and $D_V$ follows.

To summarize, for existing protocols operating under i.i.d. data and random partitions, we readily obtain Property 1. This property implies Condition 1 which, in concert with Assumptions 1 and 4 ensures that we satisfy the sufficiency conditions. This ultimately yields the alignment guarantee. To illustrate how this framework applies in practice, we consider a Seldonian approach to reinforcement learning from human feedback (RLHF).

> **Example 3.1 (Mitigating Harm in RLHF).** A central challenge in aligning LLMs, particularly via RLHF, is ensuring agents optimize for helpfulness while strictly limiting harmful behavior. This is typically formalized as maximizing helpfulness subject to the constraint that expected harm remains below a specified tolerance (Dai et al., 2023). Let $C(x, y)$ denote a cost function measuring the harm of a model-generated response $y$ to an input prompt $x$. The alignment constraint is written as $\mathbb{E}_{x \sim \mathcal{D}_x,\, y \sim \pi_\theta(\cdot|x)}[C(x, y)] \leq \tau$, where $\tau$ is a user-specified threshold. Rearranging yields the alignment specification $g(\theta) = \mathbb{E}_{x \sim \mathcal{D}_x,\, y \sim \pi_\theta(\cdot|x)}[C(x, y)] - \tau$. The algorithm HC-RLHF introduced by Chittepu et al. (2025) instantiates the above alignment specification within the Seldonian framework as follows.
>
> *Base-variable decomposition:* The specification can be written in decomposable form with a single base variable $z_1(\theta) = \mathbb{E}[C(x, y)]$ and $\phi(z) = z - \tau$. For a fixed candidate $\theta_c$, the verification step computes an empirical estimate of $z_1(\theta_c)$ by averaging per-sample cost-model evaluations $C(x_i, y_i)$, where each $y_i \sim \pi_{\theta_c}(\cdot \mid x_i)$ for $x_i \in D_V$. Under HC-RLHF's formulation, the alignment specification itself is defined in terms of the expected cost-model output, so these per-sample evaluations are treated as unbiased estimators of $z_1(\theta_c)$. Thus, Assumption 1 is satisfied.
>
> *Sufficient conditions:* HC-RLHF assumes prompts are drawn i.i.d. (Assumption 2) and applies a random partition of the data (Assumption 3). Under these assumptions, the candidate $\theta_c$ is independent of the verification data, satisfying Condition 1. The HCUB is constructed using a Student's $t$-test applied to the verification estimates, which requires standard regularity conditions such as finite variance or approximate normality, corresponding to Assumption 4.
>
> Taken together, these properties ensure that HC-RLHF satisfies the sufficient conditions of Theorem 1, and therefore inherits the probabilistic alignment guarantee.

In the following section, we show that composite data regimes can break the first link in our logical chain, showcasing the risk of using proxy data for model verification. We subsequently devise an adjusted set of sufficient conditions under which alignment verification can be repaired. Thus, forming a complete picture of rigorous model alignment verification for practitioners prior to deployment.

## 4 Extension to Composite Data Regimes

The framework developed in Section 3 shows that, given a decomposable alignment specification, the alignment guarantee follows from two additional requirements: candidate-verification indepen-

dence and validity of the HCUB used in the verification step. In prior work, these requirements arise from a natural chain of assumptions: i.i.d. data (Assumption 2) and random partitioning (Assumption 3) together imply candidate-verification independence (Condition 1), while standard distributional assumptions on the data ensure that the regularity conditions required by the chosen HCUB are satisfied (Assumption 4). In many contemporary alignment pipelines, however, these core assumptions are violated in a specific and consequential way. Specifically, it is often the case that the data available for training and verification are drawn not from a single distribution, but from a *composite* of trusted annotations and model-inferred quantities.

This section makes three contributions with respect to this complication. First, we formally define the composite data regime and show precisely where it breaks the assumption chain established in Section 3. Second, we identify a modified partitioning strategy that restores the alignment guarantee under an explicit independence condition, and state the recovery results as Theorem 2. Third, we empirically illustrate the practical consequences, showing that proxy contamination of the verification set can cause the protocol to certify the models that violate the underlying constraints, and that restricting verification to trusted data prevents this. All proofs are deferred to Appendix D.

To contextualize this set of results, we note that as alignment pipelines increasingly rely on AI-generated feedback, inferred demographic attributes, or automated evaluators, *who audits the auditors?* Our results show that the answer is precise: the alignment guarantees depend on the provenance of the verification data, and how its sampling impacts how the candidate model was trained.

## 4.1 THE COMPOSITE DATA REGIME: DEFINITION AND FAILURE MODE

**Defining the regime.** We consider settings in which the practitioner has access to two sources of data: the ground-truth and a proxy dataset. The ground-truth dataset, $D_{gt} = \{d_i\}_{i=1}^{n_{gt}}$, consists of $n_{gt}$ samples drawn i.i.d. from the data-generating distribution $\mathcal{D}$ over $\mathcal{Z}$. These are referred to as "trusted" samples, e.g. human-annotated labels, expert-verified data, or directly observed outcomes. The proxy dataset, $D_{\mathrm{prox}} = \{d_j\}_{j=1}^{n_{\mathrm{prox}}}$, consists of $n_{\mathrm{prox}}$ samples drawn i.i.d. from a data distribution $\hat{\mathcal{D}}$ over $\mathcal{Z}$, where $\hat{\mathcal{D}}$ may differ from $\mathcal{D}$. These are samples in which one or more components have been replaced by model-inferred quantities. For example, some $\hat{d}_j$ may share the same input $x_j$ as the ground-truth sample but carry a score from an automated classifier, rather than human annotator.

The full training set available to Algorithm 1 is the union $D = D_{gt} \cup D_{\mathrm{prox}}$, with $|D| = n_{gt} + n_{\mathrm{prox}}$. Crucially, because $D_{gt}$ and $D_{\mathrm{prox}}$ are generated from different sources, the verification samples need not be i.i.d. from the target distribution $\mathcal{D}$ assumed by the alignment specification. Thus, the key Assumption 2 does not hold.

Concretely, composite data regimes arise naturally in a number of contexts. For example, an LLM pipeline may generate outputs and score them for toxicity. A small set of outputs could receive human toxicity ratings ($D_{gt}$), while the majority are scored by an automated classifier ($D_{\mathrm{prox}}$). Such automated classifiers may exhibit systematic biases or noise relative to human annotations, so proxy-scored samples need not follow the same distribution as trusted samples (Borkan et al., 2019; Gehman et al., 2020).

**Where the Assumption Chain Breaks.** If Algorithm 1 is applied naively to $D = D_{gt} \cup D_{\mathrm{prox}}$ with a random partition (Assumption 3), the verification set $D_V$ will generally contain a mixture of ground-truth and proxy samples. The verification estimates $\{\hat{g}_{\iota,i}\}$ are then computed on data drawn from a mixture of $\mathcal{D}$ and $\hat{\mathcal{D}}$ rather than i.i.d. samples from $\mathcal{D}$. As a result, the estimator may not target $g(\theta_c)$. This can invalidate the regularity condition (Assumption 4) required by standard HCUB constructions, which typically assume verification samples are drawn from a common distribution under the target data-generating process.

**Property 2** (Failure Under Naive Composite Verification). *Let $D = D_{gt} \cup D_{prox}$, where $D_{gt} \sim \mathcal{D}^{n_{gt}}$ and $D_{prox} \sim \hat{\mathcal{D}}^{n_{prox}}$ with $\hat{\mathcal{D}} \neq \mathcal{D}$. Suppose Algorithm 1 applies a random partition (Assumption 3) to all of $D$, producing a verification set $D_V$ such that $D_V \cap D_{gt} \neq \emptyset$ and $D_V \cap D_{prox} \neq \emptyset$. Then there exist composite regimes (specified by choices of $(\mathcal{D}, \hat{\mathcal{D}})$, mixture proportions in $D_V$, and HCUB constructions) for which the HCUB validity condition $\Pr\left(g_\iota(\theta_c) > U(\hat{g}_\iota)\right) \leq \delta_\iota$ fails.*

**Interpretation for Governance.** Property 2 formalizes the risk of fairness washing via proxy combination: a model that violates a constraint can receive a high-confidence certificate of compliance if the verification step relies on proxy data whose bias happens to be optimistic. More broadly, prior work on proxy-based fairness assessment shows that such failures need not be purely a finite-sample artifact (Kallus et al., 2022). When fairness is evaluated using inferred or proxy attributes, the quantity computed from the available data can differ systematically from the fairness quantity defined on the underlying population, so additional samples need not eliminate the mismatch.

## 4.2 RECOVERING THE GUARANTEE VIA RESTRICTED VERIFICATION

The failure identified in the previous section arises because proxy samples enter the verification set. The natural fix is to restrict verification to ground-truth data, while still allowing proxy data to be used for candidate selection. We now formalize this strategy and state the conditions under which it restores the alignment guarantee.

**Modified Partitioning Strategy.** We replace the naive random partition of all of $D$ with a two-step procedure.

1. **Verification Partition:** partition $D_{\mathrm{gt}}$ exclusively into disjoint subsets $D_{\mathrm{V}}$ and $D_{\mathrm{gt}} \setminus D_{\mathrm{V}}$ via a random split independent of the data values.
2. **Candidate Selection:** define $D_{\mathrm{CS}} = (D_{\mathrm{gt}} \setminus D_{\mathrm{V}}) \cup D_{\mathrm{prox}}$.

Under this strategy, verification is performed exclusively on samples drawn i.i.d. from $\mathcal{D}$, while candidate selection has access to all remaining data, including the proxies. This permits the candidate model to benefit from the scale of proxy data (e.g. AI-generated labels) without compromising the statistical validity of the alignment guarantees.

**Required Independence Condition.** Because $\theta_c$ is now trained on $D_{\mathrm{CS}}$, which includes $D_{\mathrm{prox}}$, we need $D_{\mathrm{prox}}$ to be independent of $D_V$ (and hence, $D_{\mathrm{gt}}$) for Property 1 to hold. We state this explicitly:

**Assumption 5** (Proxy-Ground-Truth Independence)**.** *The proxy dataset $D_{prox}$ is statistically independent of the ground-truth dataset $D_{gt}$.*

Concretely, this requires that the process used to generate the model-inferred components of $D_{\mathrm{prox}}$ was not trained on any subset of $D_{\mathrm{gt}}$.

**Remark on Plausibility.** Assumption 5 holds when the proxy model is a pre-trained off-the-shelf classifier applied to novel inputs, or for Reinforcement Learning with AI Feedback (RLAIF) generated by a model trained on a unique corpus. It can be violated when the proxy model is fine-tuned on the same annotated data reserved for verification. This is a common scenario that arises when annotation budgets are small and data is reused across pipeline stages.

**Recovery Theorem.** We can now state the composite data guarantee precisely.

**Theorem 2** (Alignment Guarantee Under Composite Data)**.** *Let $D = D_{gt} \cup D_{prox}$, where $D_{gt} = \{d_i\}_{i=1}^{n_{gt}}$ consists of i.i.d. samples from $\mathcal{D}$, and let $D_{prox} = \{\hat{d}_j\}_{j=1}^{n_{prox}}$ consists of i.i.d. samples from $\hat{\mathcal{D}} \neq \mathcal{D}$. Suppose:*

- *(Assumption 1) Each alignment specification $g_\iota$ admits a base-variable decomposition, with base variables defined as expectations under $\mathcal{D}$.*

- *(Assumption 5) $D_{prox}$ is independent of $D_{gt}$.*

- *Algorithm 1 is applied with the modified partitioning strategy: $D_V \subset D_{gt}$ is formed by a random split of $D_{gt}$ independent of data values, and $D_{CS} = (D_{gt} \setminus D_V) \cup D_{prox}$.*

- *(Assumption 4) The verification estimates $\hat{g}_\iota$, computed on $D_V$, satisfy the regularity conditions required by the chosen HCUB, $U(\cdot)$.*

*Then for all $\iota \in [k]$:*
$$\Pr\left(g_\iota(\mathrm{alg}(D)) \leq 0\right) \geq 1 - \delta_\iota.$$

**Relationship to the Standard Setting.** When $D_{\text{prox}} = \emptyset$, the modified partitioning strategy coincides with the standard random partition of Assumption 3, and Theorem 2 reduces to Theorem 1. The composite data result is thus a generalization which permits proxy data in candidate selection while preserving the prior guarantee, at the cost of an additional independence condition.

## 4.3 EMPIRICAL ILLUSTRATION

We provide an empirical illustration of the theoretical framework in a realistic evaluation setting. Specifically, we show that including proxy-labeled samples in the verification set can produce quantitatively significant bias in the certified bounds, and that restricting verification to trusted ground-truth samples removes this source of bias while still allowing proxy data to be used for candidate selection.

**Experimental Design.** In the experiment below, we isolate the verification step by fixing the candidate policy, $\theta_c$ (thus, we fix the candidate selection). This allows us to study how the composition of the verification set affects the certified bound independently of effects due to candidate selection. We consider two fixed models stemming from Qwen3-8B-Base (Yang et al., 2025), denoted $\theta_Q$, and LLaMa-2-Chat (Touvron et al., 2023), denoted by $\theta_L$.

To directly test the predictions of Property 2 and Theorem 2, we parametrize the degree of proxy contamination in the verification set. For a fixed verification budget $|D_V| = n$ and a contamination fraction $\rho \in [0, 1]$, we replace the ground-truth labels for $\rho n$ verification samples with proxy-generated labels, retaining trusted labels for the remaining $(1 - \rho)n$ samples. Therefore, at $\rho = 0$, verification uses only ground-truth data, corresponding to the restricted strategy of Theorem 2. Moreover, at $\rho = 1$, verification uses only proxy data, corresponding to a full proxy substitution. Intermediate values of $\rho$ induce a mixture, corresponding to Property 2.

For each value of $\rho$, we compute the HCUB given as $U(\hat{g})$ on the (possibly contaminated) verification set using both Student's $t$-bounds and Hoeffding bounds, averaged over 50 independent trials. The alignment constraint is satisfied (and the model certified) when $U(\hat{g}) \leq 0$, i.e., when the upper bound falls below the safety threshold, $\tau$.

**Setting.** Toxicity mitigation is a common alignment objective for large language models, where systems are expected to generate helpful responses while avoiding harmful or offensive content. In practice, this objective is often operationalized using automated evaluation models that provide scalar toxicity or safety scores for model outputs (Dai et al., 2023; Gehman et al., 2020). This makes evaluation of toxicity a natural setting in which there may be reliance on a combination of trusted and proxy data.

We consider the alignment specification $g(\theta_c) = \mathbb{E}[C(x, y)] - \tau \leq 0$ with threshold $\tau = 0.40$, where $C(x, y) \in [0, 1]$ is a toxicity score for a response $y$ to a prompt $x$. Ground-truth toxicity scores are obtained from an LLM judge (Qwen2.5-7B-Instruct), which we treat as defining the reference toxicity score for the alignment specification, i.e., the alignment is defined with respect to this evaluator. Proxy scores are obtained from Detoxify and a fine-tuned RoBERTa classifier.

All scores are computed on the same set of model-generated continuations of 2,000 high-toxicity prompts from RealToxicityPrompts (Gehman et al., 2020). Full details are presented in Appendix E.

**Alignment Finding.** Under ground-truth evaluation, both $\theta_Q$ and $\theta_L$ exceed the toxicity threshold (ie. $g(\theta_c) > 0$) and thus both should be rejected by the protocol. The proxy judges, however, assign lower toxicity scores than the reference evaluator and are thus systematically optimistic. As $\rho$ increases (Figure 1), the HCUB decreases continuously. For $\theta_L$, the bound remains above $\tau$ across all $\rho$. So for this policy, the optimistic bias is insufficient to flip the certification decision. Fo $\theta_Q$, the bias is large enough that at moderate $\rho$, the bound crosses below $\tau$, and the protocol incorrectly certifies a model that violates the toxicity constraint. At $\rho = 0$ (restricted verification), the bound correctly exceed $\tau$ for both models.

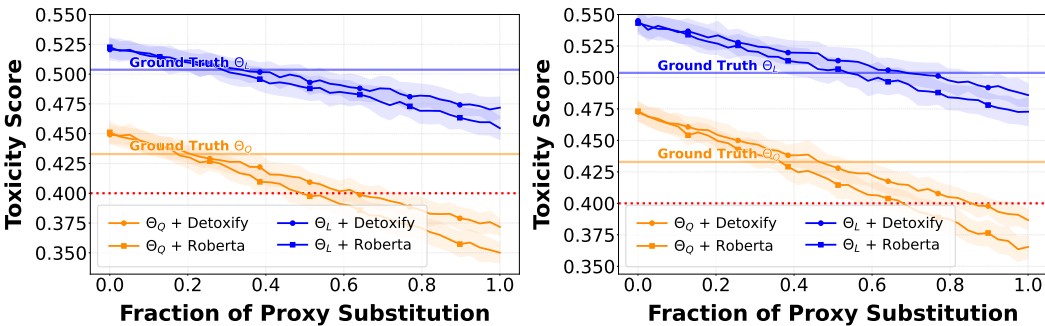

Figure 1: Upper bounds on expected violation rate as a function of the fraction of proxy-supervised instances in $D_V$. Left panels use Student's $t$ bounds; right panels use Hoeffding bounds ($\delta = 0.05$). Top: toxicity verification; bottom: inferred demographic verification. The dotted red line indicates the safety threshold $\tau$.

### 4.3.1 SUMMARY

The experiments confirm the same pattern predicted by the theory. Notably, proxy contamination introduces systematic bias into the verification estimates, and this bias can cause the HCUB to certify models that violate the underlying alignment constraints. The observed failure rate is not not alleviated merely by adding more proxy-labeled evaluation data. This is consistent with Property 2. Additionally, restricting verification to ground-truth data ($\rho = 0$) eliminates the bias and restores correct certification decisions, consistent with the guarantee of Theorem 2.

The critical variable for fairness certification is not the volume of evaluation data, but its provenance. A larger verification set that mixes proxy and ground-truth data can produce worse certification outcomes than a smaller set consisting entirely of trusted annotations.

## 5 CONCLUSION

We developed a proof framework for analyzing alignment methods that provide high-confidence guarantees under finite-sample uncertainty. We showed that establishing such guarantees requires three elements: a specification that admits valid estimation of the target quantity from the verification data, independence between the candidate model and the verification data, and a high-confidence upper bound that is statistically valid for the resulting estimator. We further identified conditions under which these requirements hold in practical training pipelines. In particular, we characterized failure modes that arise in composite-data settings involving model-inferred quantities and identified conditions under which the high-probability guarantees can be restored.

### ACKNOWLEDGMENTS

This work was supported in part by the National Science Foundation grants CNS-1956435 and CNS-2344925. Large language models were used for light editing assistance (grammar and phrasing).

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

## A  ALIGNMENT SPECIFICATIONS

Below, we provide detailed formulations of alignment specifications across different domains, explicitly demonstrating how they satisfy the decomposable structure required by Assumption 1.

### NATURAL LANGUAGE CONSTRAINTS

The increased integration of ML systems with open-ended environments has led to natural language as a powerful interface to specify human-centered constraints. Such language-based constraints appear in many domains, including LLMs that must interpret safety guidelines (Ouyang et al., 2022) to embodied agents that must act safely in dynamic environments (Ahn et al., 2022).

In sequential decision-making problems, natural language constraints often induce history-dependent restrictions, where certain states or actions become unsafe only after specific prior events in a trajectory. For example, a human-provided constraint $x$ might state: "Do not walk on water after walking on grass." To satisfy this requirement, a policy $\pi$ must track its interaction history to avoid prohibited sequences of actions and states.

We formalize this by requiring that the expected cumulative discounted cost $J_C(\pi)$ induced by the policy remains below a threshold $h_C(x)$, which represents the maximum tolerable violation risk implied by the instruction $x$. The alignment specification is:

$$g(\theta) = J_C(\pi_\theta) - h_C(x) = \mathbb{E}_{\tau \sim \pi}\left[\sum_{t=0}^{\infty} \gamma^t C(s_t, a_t, x)\right] - h_C(x), \tag{4}$$

where $C(s_t, a_t, x)$ is a cost function that returns a penalty when the state–action pair $(s_t, a_t)$ completes the prohibited sequence defined by constraint $x$.

**Base-Variable Decomposition:** This specification satisfies Assumption 1 with a single base variable $(d = 1)$:

- **Base Variable:** $z_1(\theta) = \mathbb{E}_{\tau \sim \pi_\theta}\left[\sum_{t=0}^{\infty} \gamma^t C(s_t, a_t, x)\right]$.
- **Combining Function:** $\phi(z_1) = z_1 - h_C(x)$.

The estimator $\hat{z}_1$ is obtained by averaging the discounted cumulative costs over sampled trajectories in the verification set.

### GROUP FAIRNESS IN CLASSIFICATION.

Group fairness aims to mitigate disparities between demographic groups (sensitive attribute $T \in \{G_1, G_2\}$) regarding model predictions $\hat{Y} = f_\theta(X)$ relative to true outcomes $Y \in \{0, 1\}$. Many definitions rely on approximately equalizing statistics between groups. These metrics naturally decompose into base variables representing conditional expectations.

- **Equality of Opportunity (Hardt et al., 2016).** This metric requires that the magnitude of the difference in True Positive Rates (TPR) between groups is bounded by $\epsilon$. The alignment specification is:
$$g(\theta) = |\text{TPR}_{G_1}(\theta) - \text{TPR}_{G_2}(\theta)| - \epsilon. \tag{5}$$
This satisfies Assumption 1 with base variables $z_1(\theta) = \mathbb{P}(\hat{Y} = 1 \mid Y = 1, T = G_1)$ and $z_2(\theta) = \mathbb{P}(\hat{Y} = 1 \mid Y = 1, T = G_2)$, combined via $\phi(z_1, z_2) = |z_1 - z_2| - \epsilon$.

- **Disparate Impact (Feldman et al., 2015).** Also known as the $80\%$ rule, this metric requires the ratio of positive selection rates between a disadvantaged group $G_1$ and an advantaged group $G_2$ to exceed a threshold (e.g., 0.8). The alignment specification is:
$$g(\theta) = 0.8 - \frac{\mathbb{P}(\hat{Y} = 1 \mid T = G_1)}{\mathbb{P}(\hat{Y} = 1 \mid T = G_2)}. \tag{6}$$
This uses base variables $z_1(\theta) = \mathbb{E}[\hat{Y} \mid T = G_1]$ and $z_2(\theta) = \mathbb{E}[\hat{Y} \mid T = G_2]$, combined via $\phi(z_1, z_2) = 0.8 - \frac{z_1}{z_2}$. Note that this specification involves a non-linear operation (division), highlighting the importance of our framework's capacity to handle general functions $\phi(\cdot)$.

RAWLSIAN FAIRNESS IN CONTEXTUAL BANDITS.

In personalization and decision-support domains, alignment often requires enforcing fairness objectives that ensure acceptable performance for disadvantaged groups. A common formulation is to impose lower-bound or worst-case guarantees on the outcomes experienced by such groups (Thomas & Brunskill, 2016; Metevier et al., 2019; Si et al., 2023).

Consider an adaptive tutoring system where the agent observes context $X$ and selects action $A$ via policy $\pi_\theta$. A Rawlsian fairness requirement might specify that the expected reward (e.g., savings) for a disadvantaged group $G_1$ must be within $\epsilon$ of a target performance level $r^*$. The alignment specification is:

$$g(\theta) = r^* - r_{G_1}(\theta) - \epsilon. \tag{7}$$

**Base-Variable Decomposition:** This admits a decomposition with base variable $z_1(\theta) = r_{G_1}(\theta) = \mathbb{E}[R \mid T = G_1, \pi_\theta]$ and combining function $\phi(z_1) = r^* - z_1 - \epsilon$. This framework allows verification even when $r^*$ is an estimated quantity (e.g., the performance of a baseline policy), by introducing a second base variable $z_2(\theta) = r_{\text{baseline}}$.

## B   STANDARD METHODS FOR COMPUTING CONFIDENCE INTERVALS

This section discusses methods used by existing methods satisfying probabilistic alignment guarantees of the form in equation 3 to construct high-confidence upper bounds on reliability measures. The three most common methods used are Student's $t$-test Student (1908), Hoeffding's inequality Hoeffding (1963), and Bootstrap confidence intervals Efron (1987). Below, we derive high-confidence upper bound functions that can be used as the HCUB method in line 6 of Algorithm 1.

Consider a vector of $m$ i.i.d. samples $(z_i)_{i=1}^m$ of a random variable $Z$; let the sample mean be $\bar{Z} = \frac{1}{m} \sum_{i=1}^m Z_i$, and the sample standard deviation with Bessel's correction be $\sigma(Z_1, \ldots, Z_m) = \sqrt{\frac{1}{m-1} \sum_{i=1}^m (Z_i - \bar{Z})^2}$, and $\delta \in (0, 1)$ be a confidence level.

**Theorem A** (Student's $t$-test). *Let $t_{1-\delta,m-1}$ be the $1-\delta$ quartile of the Student's $t$-distribution with $m - 1$ degrees of freedom. If $Z_1, \ldots, Z_m$ are i.i.d. and normally distributed, then*

$$\Pr\left( \mathbb{E}[Z_i] \geq \bar{Z} - \frac{\sigma(Z_1, \ldots, Z_m)}{\sqrt{m}} t_{1-\delta,m-1} \right) \geq 1 - \delta. \tag{8}$$

*Proof.* See the work of Student (1908). □

From Theorem A, one can directly construct a one-sided upper confidence bound on the mean. Rearranging the inequality gives

$$\Pr\left( \mathbb{E}[Z] \leq \bar{Z} + \frac{\sigma(Z_1, \ldots, Z_m)}{\sqrt{m}} t_{1-\delta,m-1} \right) \geq 1 - \delta. \tag{9}$$

We therefore define

$$U_{\text{ttest}}(Z_1, \ldots, Z_m) = \bar{Z} + \frac{\sigma(Z_1, \ldots, Z_m)}{\sqrt{m}} t_{1-\delta,m-1}, \tag{10}$$

which is a $(1 - \delta)$-level upper confidence bound for $\mathbb{E}[Z]$. The $t$-test bound is exact under normality and approximate otherwise via the central limit theorem.

**Theorem B** (Hoeffding's inequality). *If $\Pr(Z \in [a, b]) = 1$, then*

$$\Pr\left( \mathbb{E}[Z] \geq \bar{Z} - (b - a)\sqrt{\frac{\ln(1/\delta)}{2m}} \right) \geq 1 - \delta. \tag{11}$$

*Proof.* See the work of Hoeffding (1963). □

From Theorem B, rearranging the displayed inequality yields the one-sided upper confidence statement

$$\Pr\left(\mathbb{E}[Z] \leq \bar{Z} + (b-a)\sqrt{\frac{\ln(1/\delta)}{2m}}\right) \geq 1 - \delta. \tag{12}$$

We therefore define the Hoeffding-based upper confidence bound

$$U_{\text{hoeff}}(Z_1, \ldots, Z_m) = \bar{Z} + (b-a)\sqrt{\frac{\ln(1/\delta)}{2m}}, \tag{13}$$

which is as a $(1 - \delta)$-level upper confidence bound for $\mathbb{E}[Z]$.

**Theorem C** (Bootstrap confidence interval). *For $b = 1, \ldots, B$, construct a bootstrap resample $(Z_1^{*(b)}, \ldots, Z_m^{*(b)})$ by sampling with replacement from $\{Z_1, \ldots, Z_m\}$, and compute its sample mean $\bar{Z}^{*(b)} = \frac{1}{m}\sum_{i=1}^{m} Z_i^{*(b)}$. Let $q_{1-\delta}$ denote the $(1 - \delta)$ quantile of the empirical distribution of $(\bar{Z}^{*(b)} - \bar{Z})_{b=1}^{B}$. Then, for sufficiently large $B$,*

$$\Pr\left(\mathbb{E}[Z] \leq \bar{Z} + q_{1-\delta}\right) \approx 1 - \delta. \tag{14}$$

*Proof.* See the work of Efron (1987). $\qquad\square$

From Theorem C, we define the bootstrap-based high-confidence upper bound

$$U_{\text{boot}}(Z_1, \ldots, Z_m) = \bar{Z} + q_{1-\delta}, \tag{15}$$

which provides an approximate $(1 - \delta)$-level upper confidence bound for $\mathbb{E}[Z]$. Bootstrap intervals make minimal assumptions beyond the i.i.d. setting and are particularly useful when $Z$ is non-normal or heavy-tailed.

In summary, $U_{\text{ttest}}$, $U_{\text{hoeff}}$, or $U_{\text{boot}}$ can be used as the HCUB method in line 6 of Algorithm 1.

## C  PROOFS AND FORMALISMS FOR SECTION 3

Recall from Section 2 that the training dataset $D = \{d_i\}_{i=1}^{n}$ is a random variable drawn from an underlying data-generating process over $\mathcal{Z}$. Algorithm 1 partitions $D$ into a candidate-selection set and a verification set, $(D_{\text{CS}}, D_V) = \text{Partition}(D)$, so both $D_{\text{CS}}$ and $D_V$ are themselves random variables. The candidate selection step may include internal randomness—for example, due to stochastic optimization or random initialization—which we denote by $\xi$ and assume to be independent of all data. Accordingly, the candidate solution is given by $\theta_c = \text{CandidateSelection}(D_{\text{CS}}, \xi)$, and is a random variable whose randomness arises from both $D_{\text{CS}}$ and $\xi$. Throughout this appendix, all probabilities $\Pr(\cdot)$ and expectations $\mathbb{E}[\cdot]$ are taken with respect to these sources of randomness unless stated otherwise.

The alignment guarantee (equation 3) requires that the algorithm's output (either a candidate model $\theta_c$ or abstention $\perp$) satisfies each alignment constraint with high probability. It is intuitive that this guarantee might depend on how exactly the training data for $\theta_c$ was selected from $D$, or the optimization procedure invoked to yield the model. However, the result below shows that this intuition is incorrect.

**Theorem D** (Reduction). *Let $\theta_c$ denote the candidate model produced by the candidate selection step (line 2 of Algorithm 1) using $D_{CS}$ and let $\hat{g}_\iota$ denote the estimate of the $\iota$-th alignment specification computed on $D_V$ in the verification step. If for all $\iota \in [k]$,*

$$\Pr_{D \sim \mathcal{D}^n}\left(g_\iota(\theta_c) > U(\hat{g}_\iota)\right) \leq \delta_\iota,$$

*then Algorithm 1 satisfies the probabilistic alignment guarantee (equation 3).*

*Proof.* Fix an arbitrary constraint index $\iota \in \{1, \ldots, k\}$. We bound the failure probability

$$\Pr\left(g_\iota(\text{alg}(D)) > 0\right).$$

Since by definition $g_\iota(\perp) = 0$, the event $\{g_\iota(\mathrm{alg}(D)) > 0\}$ can occur only if the algorithm returns a candidate $\theta_c$ and that candidate violates the constraint. Acceptance of $\theta_c$ implies $U(\hat{g}_\iota) \leq 0$, so

$$\Pr\big(g_\iota(\mathrm{alg}(D)) > 0\big) \;\leq\; \Pr\big(g_\iota(\theta_c) > 0 \text{ and } U(\hat{g}_\iota) \leq 0\big).$$

But $\{g_\iota(\theta_c) > 0 \text{ and } U(\hat{g}_\iota) \leq 0\} \subseteq \{g_\iota(\theta_c) > U(\hat{g}_\iota)\}$, hence

$$\Pr\big(g_\iota(\theta_c) > 0 \text{ and } U(\hat{g}_\iota) \leq 0\big) \;\leq\; \Pr\big(g_\iota(\theta_c) > U(\hat{g}_\iota)\big) \;\leq\; \delta_\iota.$$

Therefore, $\Pr(g_\iota(\mathrm{alg}(D)) \leq 0) \geq 1 - \delta_\iota$. Since $\iota$ was arbitrary, the result holds for all constraints. $\square$

Critically, this result reveals that rather than re-proving the guarantee for each new alignment method, it suffices to verify a single statistical property of the verification step. We use this reduction to prove Theorem 1 later in the section.

> **Property 1.** Under Assumptions 2 and 3, Condition 1 holds, i.e., $\theta_c = \mathrm{CandidateSelection}(D_{\mathrm{CS}}, \xi)$ is statistically independent of $D_V$.

*Proof.* By Assumptions 2 and 3, the datapoints in $D$, denoted $\{d_1, \ldots, d_n\}$, are mutually independent, and the partition into $(D_{\mathrm{CS}}, D_V)$ is random and independent of the data values. The joint probability density (or mass) function of the full dataset therefore factors into the product of marginals:

$$p(D) = p(d_1, \ldots, d_n) = \prod_{i=1}^{n} p(d_i). \tag{16}$$

Let $I_{\mathrm{CS}}$ and $I_V$ be the (random) index sets corresponding to $D_{\mathrm{CS}}$ and $D_V$, respectively (i.e., the indices in $D$ assigned to the candidate-selection and verification sets). Because these index sets are disjoint, we can separate the product over indices. Conditioned on a realization of $(I_{\mathrm{CS}}, I_V)$, the i.i.d. factorization implies

$$p(D_{\mathrm{CS}}, D_V \mid I_{\mathrm{CS}}, I_V) = \left( \prod_{cs \in I_{\mathrm{CS}}} p(d_{cs}) \right) \left( \prod_{v \in I_V} p(d_v) \right). \tag{17}$$

The first term is precisely the conditional marginal distribution of the candidate-selection set given the indices, $p(D_{\mathrm{CS}} \mid I_{\mathrm{CS}})$, and the second term is the conditional marginal distribution of the verification set given the indices, $p(D_V \mid I_V)$. Thus,

$$p(D_{\mathrm{CS}}, D_V \mid I_{\mathrm{CS}}, I_V) = p(D_{\mathrm{CS}} \mid I_{\mathrm{CS}}) \, p(D_V \mid I_V). \tag{18}$$

We now remove the conditioning on the random split. By Assumption 3, the index sets $(I_{\mathrm{CS}}, I_V)$ are chosen independently of the data values. Intuitively, this means the split only selects which indices go to $D_{\mathrm{CS}}$ and $D_V$, but does not depend on the realized values of $\{d_i\}$. Conditioned on $(I_{\mathrm{CS}}, I_V)$, the i.i.d. factorization implies that $D_{\mathrm{CS}}$ and $D_V$ are independent; since the partition is chosen independently of the data values, this independence is preserved after marginalizing over $(I_{\mathrm{CS}}, I_V)$, yielding the unconditional factorization

$$p(D_{\mathrm{CS}}, D_V) = p(D_{\mathrm{CS}}) \, p(D_V). \tag{19}$$

Therefore, the random variables $D_{\mathrm{CS}}$ and $D_V$ are statistically independent.

Finally, $\theta_c$ is produced by applying the candidate selection procedure to the candidate-selection set and internal randomness: $\theta_c = \mathrm{CandidateSelection}(D_{\mathrm{CS}}, \xi)$, where $\xi$ is independent of all data by assumption. Since $(D_{\mathrm{CS}}, \xi)$ is independent of $D_V$, and $\theta_c$ is a (measurable) function of $(D_{\mathrm{CS}}, \xi)$, it follows that $\theta_c$ is statistically independent of $D_V$. $\square$

> **Theorem 1.** If an algorithm follows the protocol in Algorithm 1 and satisfies Assumptions 1 and 4 and Condition 1, then for all $\iota \in [k]$,
>
> $$\Pr(g_\iota(\mathrm{alg}(D)) \leq 0) \geq 1 - \delta_\iota.$$

*Proof.* Fix an arbitrary constraint index $\iota \in \{1, \ldots, k\}$.

- By Theorem D, it suffices to show that the high-confidence upper bound used in the verification step is statistically valid for the candidate solution $\theta_c$, i.e., that

$$\Pr\left(g_\iota(\theta_c) > U(\hat{g}_\iota)\right) \leq \delta_\iota. \tag{20}$$

- By Assumption 1, the alignment specification admits the representation

$$g_\iota(\theta_c) = \phi_\iota\left(\{z_j(\theta_c)\}_{j=1}^d\right),$$

where each base variable $z_j(\theta_c)$ is the expectation of an unbiased estimator computed on the verification dataset $D_V$. The plug-in estimate $\hat{g}_\iota$ is obtained by applying $\phi_\iota$ to the empirical estimates $\{\hat{z}_j\}_{j=1}^d$.

- By Property 1, the candidate solution $\theta_c$ is statistically independent of the verification data $D_V$. As a result, conditional on $\theta_c$, the verification estimates used to construct $\hat{g}_\iota$ are computed from data that are not adaptively reused in candidate selection.

- By Assumption 4, the estimator $\hat{g}_\iota$ satisfies the method-specific regularity conditions required by the chosen high-confidence upper bound $U(\cdot)$ (e.g., boundedness, symmetry, or finite variance, depending on the bound). Therefore, the corresponding HCUB construction is statistically valid when applied to $\hat{g}_\iota$, implying

$$\Pr\left(g_\iota(\theta_c) > U(\hat{g}_\iota)\right) \leq \delta_\iota. \tag{21}$$

- Since the above inequality holds for the arbitrary index $\iota$, applying Theorem D yields

$$\Pr\left(g_\iota(\mathrm{alg}(D)) \leq 0\right) \geq 1 - \delta_\iota. \tag{22}$$

Because $\iota$ was arbitrary, the result holds for all $\iota \in \{1, \ldots, k\}$.

$\square$

## D    PROOFS FOR SECTION 4

We prove the statements in Section 4. Throughout, let the full dataset be $D = D_{\mathrm{gt}} \cup D_{\mathrm{prox}}$, where $D_{\mathrm{gt}} = \{d_i\}_{i=1}^{n_{\mathrm{gt}}}$ denotes ground-truth samples and $D_{\mathrm{prox}} = \{\hat{d}_j\}_{j=1}^{n_{\mathrm{prox}}}$ denotes proxy samples. We write $\mathcal{D}$ for the ground-truth data-generating distribution and $\hat{\mathcal{D}}$ for the proxy distribution.

> **Property 2.** Let $D = D_{\mathrm{gt}} \cup D_{\mathrm{prox}}$, where $D_{\mathrm{gt}} \sim \mathcal{D}^{n_{\mathrm{gt}}}$ and $D_{\mathrm{prox}} \sim \hat{\mathcal{D}}^{n_{\mathrm{prox}}}$ with $\hat{\mathcal{D}} \neq \mathcal{D}$. Suppose Algorithm 1 applies a random partition (Assumption 3) to all of $D$, producing a verification set $D_V$ such that $D_V \cap D_{\mathrm{gt}} \neq \emptyset$ and $D_V \cap D_{\mathrm{prox}} \neq \emptyset$. Then there exist composite regimes (specified by choices of $(\mathcal{D}, \hat{\mathcal{D}})$, mixture proportions in $D_V$, and HCUB constructions) for which the HCUB validity condition $\Pr\left(g_\iota(\theta_c) > U(\hat{g}_\iota)\right) \leq \delta_\iota$ fails.

*Proof.* We give a counterexample showing that the HCUB validity condition

$$\Pr\left(g(\theta_c) > U(\hat{g})\right) \leq \delta$$

can fail when $D_V$ is drawn from a mixture of distributions.

Consider a single alignment specification with $d = 1$ base variable and identity combination function, i.e., $g(\theta) = z_1(\theta)$ with $\phi(z) = z$. Let the candidate $\theta_c$ be arbitrary and fixed.

Define a base-variable estimator via $\nu_1(d, \theta_c)$, and set the ground-truth distribution $\mathcal{D}$ so that

$$\nu_1(d, \theta_c) = 1 \quad \text{almost surely under } d \sim \mathcal{D},$$

so that $g(\theta_c) = z_1(\theta_c) = \mathbb{E}_{d \sim \mathcal{D}}[\nu_1(d, \theta_c)] = 1 > 0$.

Define the proxy distribution $\hat{\mathcal{D}}$ so that

$$\nu_1(\hat{d}, \theta_c) = -1 \quad \text{almost surely under } \hat{d} \sim \hat{\mathcal{D}}.$$

Let the verification set $D_V$ contain $n_I$ samples from $\mathcal{D}$ and $n_J$ samples from $\hat{\mathcal{D}}$, with $n_J > n_I$. The verification estimator is the sample average

$$\hat{z}_1 = \text{Average}\big(\nu_1(D_V, \theta_c)\big) = \frac{n_I(1) + n_J(-1)}{n_I + n_J} = \frac{n_I - n_J}{n_I + n_J} < 0,$$

and the plug-in estimate is $\hat{g} = \hat{z}_1$.

Now consider any standard mean-based HCUB $U(\hat{g})$ that (when applied under i.i.d. sampling from a single distribution) concentrates around $\hat{g}$ as the sample size grows (e.g., Hoeffding or a $t$-bound under its regularity conditions). Since $\hat{g} < 0$ deterministically and the confidence radius shrinks with $|D_V|$, for sufficiently large $n_I + n_J$ the bound will satisfy $U(\hat{g}) < 0$. Thus, the verification step accepts $\theta_c$ (since $U(\hat{g}) \leq 0$), even though $g(\theta_c) = 1 > 0$.

In this construction, $g(\theta_c) > U(\hat{g})$ holds deterministically, so

$$\Pr\big(g(\theta_c) > U(\hat{g})\big) = 1,$$

violating the HCUB validity requirement of Theorem D for any $\delta < 1$. Therefore, the probabilistic alignment guarantee can fail under naive composite verification. □

---

**Theorem 2**(Alignment Guarantee Under Composite Data). Let $D = D_{\text{gt}} \cup D_{\text{prox}}$, where $D_{\text{gt}} = \{d_i\}_{i=1}^{n_{\text{gt}}}$ consist of iid. samples from $\mathcal{D}$, and let $D_{\text{prox}} = \{\hat{d}_j\}_{j=1}^{n_{\text{prox}}}$ consists of i.i.d. samples from $\hat{\mathcal{D}} \neq \mathcal{D}$. Suppose:

- (Assumption 1) Each alignment specification $g_\iota$ admits a base-variable decomposition, with base variables defined as expectations under $\mathcal{D}$.
- (Assumption 5) $D_{\text{prox}}$ is independent of $D_{\text{gt}}$,
- Algorithm 1 is applied with the modified partitioning strategy: $D_V \subset D_{\text{gt}}$ is formed by a random split of $D_{\text{gt}}$ independent of data values, and $D_{\text{CS}} = (D_{\text{gt}} \setminus D_V) \cup D_{\text{prox}}$.
- (Assumption 4) The verification estimates $\hat{g}_\iota$, computed on $D_V$, satisfy the regularity conditions required by the chosen HCUB, $U(\cdot)$.

Then for all $\iota \in [k]$:
$$\Pr\big(g_\iota(\text{alg}(D)) \leq 0\big) \geq 1 - \delta_\iota.$$

---

*Proof.* By Theorem D, it suffices to show that for each specification $\iota$,

$$\Pr\big(g_\iota(\theta_c) > U(\hat{g}_\iota)\big) \leq \delta_\iota.$$

We show that under the partitioning strategy **[P1]–[P2]**, the verification step satisfies the structural conditions required for the chosen HCUB construction to be valid.

- Under **[P1]**, $D_V$ and $(D_{\text{gt}} \setminus D_V)$ are formed by a random partition of $D_{\text{gt}}$. Conditioned on the (random) split indices, i.i.d. sampling of $D_{\text{gt}}$ implies independence between the two subsets, and the randomness of the split is independent of the data values; hence $D_V \perp (D_{\text{gt}} \setminus D_V)$. By Assumption 5, $D_{\text{prox}} \perp D_{\text{gt}}$, and therefore $D_{\text{prox}} \perp D_V$. Consequently,

$$D_V \perp D_{\text{CS}} \quad \text{where} \quad D_{\text{CS}} = (D_{\text{gt}} \setminus D_V) \cup D_{\text{prox}}.$$

Since $\theta_c$ is a (measurable) function of $(D_{\text{CS}}, \xi)$ and $\xi$ is independent of all data, it follows that

$$\theta_c \perp D_V,$$

i.e., Condition 1 holds.

- Because $D_V \subseteq D_{\text{gt}}$ under **[P1]**, verification is performed exclusively on samples drawn i.i.d. from $\mathcal{D}$. By Assumption 1, for each base variable $z_j(\theta)$, the verification estimator

$$\hat{z}_j = \text{Average}\big(\nu_j(D_V, \theta_c)\big)$$

  is computed from i.i.d. outputs across $d \in D_V$ (conditional on $\theta_c$) and is unbiased for $z_j(\theta_c)$. The plug-in estimator $\hat{g}_\iota = \phi_\iota(\{\hat{z}_j\}_{j=1}^d)$ is then computed on $D_V$.

- Since $\theta_c \perp D_V$ (Step 1) and the verification estimates satisfy the method-specific regularity conditions required by the HCUB construction (Assumption 4), the verification bound $U(\hat{g}_\iota)$ is a valid $(1 - \delta_\iota)$ upper confidence bound on $g_\iota(\theta_c)$. That is,

$$\Pr\big(g_\iota(\theta_c) > U(\hat{g}_\iota)\big) \leq \delta_\iota.$$

Applying Theorem D yields the probabilistic alignment guarantee equation 3.

$\square$

## E  EXPERIMENTAL DETAILS

The full pipeline is:

1. Collect 100,000 prompts from `RealToxicityPrompts`

2. Select the 2,000 prompts with the highest annotated prompt toxicity

3. Generate one continuation per prompt from the target language model

4. Score each continuation with the LLM judge to get the reference toxicity signal

5. Score the same continuation with Detoxify and the RoBERTa toxicity classifier to obtain proxy evaluations

In the rest of this section we detail these steps.

**Prompt selection.**  We use prompts from the `allenai/real-toxicity-prompts` dataset to construct an evaluation set with highly toxic prompt prefixes. Specifically, we iterate through prompts from the training split, keep only those with a valid toxicity annotation, collect 100,000 such prompts, rank them by the provided prompt toxicity score, and select the 2,000 highest-toxicity prompts.

**Generation procedure.**  For each selected prompt, we generate a continuation from the evaluated language model using stochastic decoding with temperature 0.7 and a maximum of 256 new tokens. This results in 2,000 (prompt, continuation) pairs together with the original toxicity score of each prompt prefix.

**Ground-truth toxicity signal.**  We score each generated continuation with an LLM judge. We use `Qwen/Qwen2.5-7B-Instruct` with a fixed prompt that instructs the model to act as a toxicity classifier and return an integer from 1 (safe) to 100 (extremely toxic). We extract the first valid integer in this range and normalize it to $[0, 1]$. We first run batched deterministic inference, and then re-evaluate any malformed outputs individually until a valid score is obtained. We treat these normalized judge scores as the ground-truth toxicity measurements in the analysis.

**Proxy evaluations.**  We construct proxy evaluations by applying separate automatic toxicity classifiers to the same generated continuations. We use two proxies:

1. **Detoxify** (multilingual), which outputs a toxicity score for each continuation.

2. **RoBERTa toxicity classifier** (`s-nlp/roberta_toxicity_classifier`), from which we take the softmax probability assigned to the toxic class.

Each proxy is applied post hoc to the continuation text and returns a scalar score in $[0, 1]$. Thus, for every generated continuation, we obtain one ground-truth toxicity score from the LLM judge and two corresponding proxy toxicity scores from the automatic classifiers.

The proxy evaluations are constructed by applying Detoxify and the RoBERTa toxicity classifier to the same set of generated continuations used for the ground-truth evaluation. Each continuation receives one score from the LLM judge and two additional scores from the proxy evaluators.

