# OpenReview forum: "Verifying Alignment Constraints Under Finite-Sample Uncertainty in Composite-Data Regimes"
_ICLR.cc/2026/Workshop/AFAA — AFAA 2026 Poster_

### Official Review · Reviewer_u5Ss · 2026-02-16
**Statistical certification for fairness can fail under proxy-based evaluation: a clear analysis of composite-data risks with practical conditions for reliable verification**

**Rating:** 4
**Confidence:** 4

**Summary:**

This paper studies “certification-based” alignment protocols that aim to provide high-confidence guarantees that an aligned system satisfies fairness/safety constraints under statistical uncertainty. The authors formalize a common three-step protocol: data partition, candidate selection, and verification, and prove a clean reduction showing that end-to-end probabilistic alignment guarantees hinge on the validity of the verification bound (HCUB) rather than the specific optimization used to obtain the candidate solution (Theorem 1).

The core technical contribution is then an analysis of how these guarantees can fail in “composite data regimes” common in modern alignment (e.g., RLAIF), where verification uses a mixture of ground-truth and proxy evaluations, breaking assumptions behind standard confidence bounds. The paper formalizes this failure (Lemma 1) and proposes a sufficient-condition fix: restricted verification using only ground-truth samples, plus an independence condition between proxy generation and the ground-truth set (Assumption 5 / Theorem 3).

Empirically, the authors illustrate the predicted failure mode by varying the fraction of proxy substitutions in the verification set in two settings: (i) toxicity verification where proxy judges are optimistic relative to an LLM-as-judge “ground truth,” and (ii) inferred demographic attributes in a “minor safety” auditing setting, showing that proxy contamination can spuriously lower upper bounds and flip accept/reject decisions.

**Strengths:**

1. Clear conceptual reduction that unifies many protocols. Theorem 1 isolates where statistical validity is won or lost at verification, making the paper’s message easy to transfer across RLHF/RLAIF/prompting contexts.

2. Timely problem formulation for modern alignment pipelines. The composite-data framing matches real practice (proxy judges, inferred attributes, adaptive evaluation loops), and the paper explicitly connects these to broken i.i.d./independence assumptions.

3. Concrete failure mode + principled recovery condition. Lemma 1 and Theorem 3 give a crisp “what fails / how to fix” story that a practitioner can operationalize (verify on ground-truth only; don’t reuse ground-truth to build the proxy).

4. Empirical illustration aligns with theory. The proxy-substitution sweep is a simple design that directly tests the paper’s main claim: more proxy contamination → lower estimated bounds → greater risk of false certification.

**Weaknesses:**

1. “Ground truth” in experiments is still model-mediated. In toxicity, the paper treats an LLM-as-judge (Qwen2.5-7B-Instruct) as ground-truth while other learned judges are “proxy.” This is reasonable for a controlled study, but it blurs the normative message: if the “ground truth” judge is itself imperfect or correlated with proxies, the empirical conclusions may be sensitive.

2. Limited exploration of dependency structures beyond mixture contamination. The paper discusses other realistic couplings (shared calibration sets, overlapping representations, adaptive evaluation), but the experiments primarily vary proxy fraction ρ. A broader empirical stress test (e.g., correlated proxy/ground-truth errors, shared training overlap) would strengthen the claim that the issue persists beyond simple optimism bias.

3. Trade-off is acknowledged but not quantified. The conclusion highlights the robustness–efficiency trade-off (restricted verification widens bounds, increases abstention), but the empirical section as shown does not deeply quantify abstention rates vs. verification budget vs. proxy usage, likely important for deployment decisions.

4. Assumption clarity could be improved for practitioners. Assumption 5 (proxy–ground-truth independence) is central, but in real pipelines proxies are often trained/tuned on overlapping human labels. A short “audit checklist” or diagnostic guidance would make the theoretical condition easier to interpret operationally.






Suggestions for improvement:
1. Add an experiment where the “proxy” judge is partially trained on (or calibrated using) the same ground-truth set, to demonstrate the kind of dependence the paper flags as a key open problem.

2. Report abstention/NSF frequency (or implied accept/reject frequency) as a function of verification budget n and δ to make the robustness–efficiency trade-off concrete.

3. Provide a short “practitioner” subsection translating assumptions into pipeline guidance: how to structure splits, how to avoid judge reuse, and what failure signatures might look like.

---

### Official Review · Reviewer_ijGV · 2026-02-20
**Better Evaluation of Alignment**

**Rating:** 3
**Confidence:** 2

**Summary:**

The paper presents a theoretical framework to break down the alignment process. It identifies common structures in alignment protocols and the authors show how the alignment pipeline's evaluation can be reduced to the final evaluation / verification step and in particular the identification of a high confidence upper bound. They demonstrate how alignment evaluation fails if inadequate data is used for evaluation using a case study of toxicity verification and include another case study displaying failure modes when inferring a demographic attribute. They also show how, even if a model is trained on mixed or proxy feedback (e.g. RLAIF), the statistical validity of evaluation is possible as long as the evaluation happens on appropriate i.e. ground truth data.

**Strengths:**

The paper contains a strong theoretical underpinning for its framework including proofs for many of its claims. It highlights realistic issues which may affect current evaluation pipelines. The paper further includes empirical analyses to demonstrate and validate some of the derived takeaways from the theoretical framework. The work closes with an interesting discussion highlighting potentially valuable avenues for future work.

**Weaknesses:**

- I have found this paper difficult to follow and I believe it would have benefitted from a clearer description of its main contribution(s), especially a clear and concise description of its framework and how to utilize it.
- The paper mentions agentic systems and RLAIF many times, yet does not strongly demonstrate the applicability of the framework in the context of agentic systems. Including an empirical experiment within an agentic context seems appropriate when "Agentic Alignment" is literally in the title of the paper.
- While the framework is an interesting contribution, the two takeaways deduced and empirically examined are not very novel or surprising. The fact that proxy substitution adversely affects the results / statistical validity of an (alignment) evaluation is to be expected. A more interesting take-away would be, if the framework could help with the design of more robust evaluation protocols under suboptimal conditions or to provide a bounding of the estimated upper bonds even under proxy substitution.
- In that light, the paper would benefit from a clear description of how the framework is best to be used and the next steps. While the authors mention, the importance of "[...] verification procedures that safely incorporate heterogeneous or proxy supervision without sacrificing statistical guarantees," it remains unclear whether their framework could be helpful in this light.
- The paper does not include code for its empirical experiments. I do not see a compelling reason why not and would strongly urge the authors to release their experiment and analysis code for the sake of reproducibility.
- The empirical experiments in the paper are not sufficiently clear described.
	- For the toxicity experiment, it is not perfectly clear how proxy evaluations were created in L392. The description here could be expanded.
	- For the inferred demographic personas descriptions are very vague as to how prompts have been created and whether this procedure is grounded in prior work.
	- The two empirical experiments seem rather arbitrary in their creation and setup. Referencing prior work utilising similar experiments would help to make these more convincing.
- As far as I know, this work does not include a statement on the usage of LLMs. I understand this as corresponding to "no LLMs being used at all during its creation". I would ask the authors to explicitly state whether this has been the case. If LLMs were used at all they are required to transparently disclose this as part of the workshop's / conference's LLM policy.
- L049, L050 references contain double brackets.
- L618: The reference for Weber et al from NeurIPS is lacking a year, the same seems to be the case for Tse et al (L613). This suggests a strong need to review the correctness of references.
- L0430 typos "h igh-risk" and "interactiosn"

---

### Official Review · Reviewer_5uzA · 2026-02-21
**Review for submission 25**

**Rating:** 3
**Confidence:** 2

**Summary:**

This paper studies how to statistically verify fairness/alignment guarantees in LLM/agent pipelines under finite data, shows that standard guarantees can fail in composite/proxy-data settings (e.g., RLAIF), and proposes conditions under which valid guarantees can be recovered.

Disclaimer: I am not an expert in the related field, so my judgment is very limited.

**Strengths:**

The problem is important and timely: the paper addresses statistical verification of fairness/safety constraints in modern alignment pipelines, where empirical evaluation alone may fail to provide reliable guarantees under finite data and proxy supervision. The motivation around statistical coupling, fairness washing, and composite data regimes (e.g., RLAIF) is compelling and practically relevant.

**Weaknesses:**

The main weakness is that the paper is difficult to read and follow, despite addressing an important topic.

1. The paper introduces many symbols and abstractions very early on, which makes it hard to build intuition before the formal framework is fully established.
2. The logical chain (Assumptions 1/4 + Property 2 → Theorem 2; then composite-data violations + Assumption 5 + restricted partitioning → Theorem 3) is valid, but the presentation requires substantial back-and-forth to track what each condition is doing and where it is used
3. There are visible typo/formatting issues (e.g., “Toxisity Score” in Figure 1, and citation formatting inconsistencies in references.

---

### Meta-Review · Area_Chair_Kcg7 · 2026-02-23

**Recommendation:** Main Papers Track
**Confidence:** 4

**Metareview:**

All reviewers agree that the paper has important contributions, is technically solid, and has a good motivation. I strongly encourage the authors to incorporate the reviewers' suggestions, in particular:
- improve presentation quality and logical train of thought between technical results
- explain evaluation criteria better in the experimental section and motivate the experimental design
- clarify the way 'ground truth' is operationalized in the paper in order to avoid misleading readers.

I recommend the paper be accepted as a poster.

---

### Decision · Program_Chairs · 2026-03-02

Accept (Poster)